# First Interaction Network of Sarcosaprophagous Fauna (Acari and Insecta) Associated with Animal Remains in a Mediterranean Region (Northern Spain)

**DOI:** 10.3390/insects13070610

**Published:** 2022-07-06

**Authors:** Sandra Pérez-Martínez, María Lourdes Moraza

**Affiliations:** Department of Environmental Biology, University of Navarra, 31008 Pamplona, Navarra, Spain; mlmoraza@unav.es

**Keywords:** interaction network, mites, beetles, flies, phoresy, Iberian Peninsula, Mediterranean region, Navarra

## Abstract

**Simple Summary:**

Forensic entomology applies the knowledge of arthropods to obtain useful information for the resolution of forensic investigations. In legal medicine, one of the methods used for the dating of death is the faunal succession, which is based on the orderly and predictable changes of the species associated with a corpse over time. The potential of insects for this purpose has been known for more than seven centuries, but mites are only currently being incorporated. Mites can provide useful information on the time and place of death because: they are a very diverse group, ubiquitous, abundant, and frequent; they contribute to the faunal succession; they are indicators of location and transfer of the corpse; they can be present in unfavorable conditions or environments for insects; they complement the information obtained from insects; and they are tracking evidence. Due to the absence of knowledge regarding the sarcosaprophagous fauna in the Mediterranean region, it is necessary to highlight the diversity of insects, together with their phoretic mites, present in decomposing animal organic matter, and their interaction network in this region, in order that the results obtained can be applied in future forensic investigations and contribute to the estimation of the time and place of death.

**Abstract:**

The potential of insects for forensic investigations has been known for more than 700 years. However, arthropods such as mites could also play a role in these investigations. The information obtained from insects, together with their phoretic mites, is of special interest in terms of estimating the time and geographical location of death. This paper presents the first interaction network between phoretic mites and their host insects in Navarra. It also reports the first time that an interaction network was applied to animal remains of forensic relevance. The data reveal the degrees of specificity of the interactions established, the biological and ecological characteristics of the mites at the time of association, and factors that played important roles in the mites’ dispersion. Fauna was collected using 657 traps baited with 20 g of pig carrion over a year. Only 0.6% of insects collected carried phoretic mites. The network comprised 312 insects (275 beetles, 37 flies) and 1533 mites and was analyzed using various packages of the R programming language. We contribute new host insect records for 15 mites, 3 new records of insects as hosts, 5 new mite records for the Iberian Peninsula, and 2 new mites records and 8 new insect records for Navarra.

## 1. Introduction

Forensic entomology involves the application of the knowledge of arthropods to forensic investigations to obtain information useful for the resolution of cases that will be heard in the courts of justice [1,2,3,4,5,6].

Corpses, human or otherwise, are temporary sources of resources exploited by many arthropods. Insects and mites are the most numerous, diverse, and persistent [1]. The information obtained on both groups, individually or together, can be useful in forensic investigations [1,2,7]. Insects and their phoretic mites can aid in the calculation of the time and place of death; they constitute one of the most accurate dating methods (they reach the corpses within a few minutes or hours of death, depending on the season and environmental conditions) [1,8,9,10,11,12], and even the only valid dating method after the first 72 h [13,14]. In addition, using mites, it can be determined whether the corpse has been moved or manipulated [4,15,16,17,18].

One of the methods currently used to estimate the time of death, post-mortem interval (PMI), is faunal succession. This method estimates the PMI based on the type and composition of species found in the cadaveric environment because the fauna arrives in orderly successive waves throughout the decomposition process [11,17,19], and is, as such, predictable and indicative [19,20].

Traditionally, the arrival of mites has been associated with the end of the decomposition process because they are overlooked until the larval mass of insects abandons the body [12,20,21]. However, they are present from the beginning of the decomposition process, arriving either by themselves or attached to necrophagous and necrophiles insects which visit the corpse [22,23]. 

Mites’ dispersal is essential for the survival of the species [24] that live in discontinuous, unpredictable habitats or in temporary environments (such as corpses) where resources are finite [25]. Mites have developed the strategy of phoresy to be transported by other highly mobile animals from a depleted source to another to be exploited, and where it or its progeny can prosper [26,27].

In corpses, mites that have arrived as phoretic are the most abundant. Despite an absence of detailed studies of the direct mite–insect relationships, more than 200 species have been cited associated with flies and beetles [28]. The dispersive stage is the most resistant and longest lived, and it is specific to each family [29]. Immature stages (larva, protonymph and/or deutonymph) and adults (males and females) in Mesostigmata mites or hypopus (deutonymph highly adapted to phoresy) of Astigmata can travel. Generally, mesostigmatid females or deutonymphs are the phoretic stage [24,30], although occasionally males have been found [25]. Throughout their evolutionary history, phoretic mites have undergone several morphological, physiological, and behavioral changes that allowed them to adapt to their hosts, favoring their dispersal [31,32]. Some of these adaptations involve changes in body structures, synchronization of their life cycles with that of their host, specificity and location on the carrier, and attachment mechanisms [12,25].

The potential role of insects (especially flies and beetles) in forensic investigations has been known for more than 700 years. However, arthropods such as mites have been frequently ignored due to their size, the difficulty of identifying them, and ignorance of their biology [4,12]. At present, there is a move to incorporate them in forensic investigations because their high diversity, ubiquity, abundance, and frequency. As they contribute to faunal succession, they are useful in estimating the PMI, they are indicators of location and movement of the corpse, they can be present in conditions or environments unfavorable for insects, they complement the information obtained from the insects, and they are evidence of tracking [22,33,34,35,36].

For all these reasons, and the shortage of knowledge regarding the relations established between the sarcosaprophagous fauna in the Mediterranean region, it was necessary to construct an interaction network made up of mites and their host insects (flies and beetles) in the decomposition of organic animal matter.

## 2. Materials and Methods

***Sample collection and identification.*** Fieldwork was carried out in the zone called “El Carrascal” (Unzué) (42.651411, −1.641214) located in the middle eastern region of the Foral Community of Navarra (northern Spain) and previously described in Pérez-Martinez *et al.*’s 2019 paper [34]. The vegetation in this area remains green all year round, although it may experience periods of drought. The climate is Mediterranean with cool summers [37]. A sampling area of 600 m^2^ was chosen for its characteristics: it is isolated and far from urban centers, and because cadavers are likely found in roadside ditches, near motorways (AP-15) and on country roads (NA-121). Traps used to collect faunistic samples and their distribution in the research area have been previously described in Pérez-Martinez *et al.* 2019 [34]. A total of 657 samples were analyzed, from 247 McPhail modified traps, 246 pitfall modified traps, and 164 carrion on-surface traps. The bait used was 20 g of a mixture of heart, lung, and liver of *Sus scrofa domestica* L., since this species is considered a model of human decomposition for its anatomical and nutritional similarities [1,7,38,39]. The preservative used was 30% propylene glycol. The sampling took place from 7 April 2017 to 5 April 2018. Traps were revised and bait replaced each week. The content of each trap was kept in glass jars with 70% alcohol.

In the laboratory, insect (infected and non-infected) and free mites were sorted and quantified. Infected insects were examined to obtain the following information about each of its phoretic mites: position(s) on the insect body, number of individuals, stage(s), sex(es), and attachment mechanism(s). Finally, everyone was identified (312 insects and 1533 mites). Mesostigmata mites were cleared using Nesbitt’s fluid, mounted in Hoyer’s medium on microscope slides, and identified under a microscope equipped with a phase contrast optical system (OLYMPUS OPTICAL CO., LTD. model BX51TF, Tokyo, Japan) following the specialized taxonomic keys [30,40,41,42,43,44,45,46,47,48,49,50,51,52,53,54]. Astigmata mites were preserved in alcohol 70% for future identification. Host insects were identified under a magnifying glass (Nikon, model C-LEDS, Shanghai, China). Beetles of the family Scarabaeidae were identified by Dr. Pablo Bahillo de la Puebla, and those of the family Staphylinidae by Dr. Raimundo Outerelo Domínguez. Some individuals could not be identified to the species level, so they were treated as if they belonged to the same species and registered with the name of the family.

***Data analysis*.** The interaction network “phoretic mites-host insects” and biodiversity and specificity indexes for each species were obtained using various packages of the R programming language, version 3.6.3 [55].

The bipartite package by Dormann *et al.* [56] includes functions that allow calculating and visualizing different indices for the description of two-level ecological networks [57,58]. These connections are established between species of two groups (insects and mites) [59,60]. We used *Plotweb* (draw a bipartite graph from a two-dimensional matrix), *Visweb* (converts a two-dimensional matrix to a grid representation of the bipartite network), *Networklevel* (computes indices and values of a bipartite network), and *Specieslevel* (calculates specific indices of a network and describes its participants). Network indexes were determined as established by their authors. *Links by species* (the average number of interactions by species)*, network asymmetry* (balance between the number of individuals at both levels; positive values indicate higher numbers in the low level (insects), negative values indicate higher numbers in the high level (mites), and values close to 0 indicate an equilibrium between both levels); *number of compartments* (subsets of the network that are not connected to the rest of the compartments); *interaction uniformity (H_2’_)* (it is a measure of network specialization). The underlying equation is Shannon’s (H_2_), but the value has been computed for the given network and standardized with the minimum H_2_ and the maximum H_2_. H_2_’ values range from 0 (no specialization) to 1 (perfect specialization) for *species specificity index* (degree of specificity that one species has towards another; values close to 0 indicate low specificity, and values close to 1 indicate high specificity). Simpson’s biodiversity and specificity indexes were calculated for each mite species. The *Simpson index* indicates the probability that two randomly selected individuals, occupying the same habitat, belong to the same species. Values close to 0 indicate high probability of dominance of a species, whereas values close to 1 indicate habitats of greater diversity.

The insects and mites’ specimens were deposited in the Museum of Zoology, University of Navarra (MZUNAV), Pamplona, Spain.

## 3. Results

A total of 51909 arthropods were captured in this study: 93% insects (41105 flies and 6987 beetles) and 7% mites (3817); 0.6% of insects (312 specimens) carried phoretic mites: 88% beetles (275) and 12% flies (37). These insects belonged to 13 families, 19 genera, and 24 species. Calliphoridae: *Calliphora vomitoria* (Linnaeus, 1758)*, Chrysomya albiceps* (Wiedemann, 1819), *Chrysomya megacephala*** (Fabricius, 1794)*, Lucilia silvarum*** (Meigen, 1826)*;* Dermestidae: *Dermestes frischii*** Kugelann, 1792, *Dermestes undulatus*** Brahm, 1790; Geotrupidae: *Geotrupes*
*(Geotrupes) mutator* (Marsham, 1802); Histeridae: *Hister unicolor* Linnaeus, 1758, *Margarinotus brunneus*** (Fabricius, 1775)*, Saprinus*
*(Saprinus) subnitescens*** Bickhardt, 1909; Muscidae: *Hydrotaea* sp., *Neomyia* sp.; Scarabaeidae: *Onthophagus (Paleonthophagus)*
*coenobita* (Herbst, 1783); Silphidae: *Nicrophorus interruptus* (Stephens, 1830), *Nicrophorus vespilloides* Herbst, 1783, *Thanatophilus ruficornis* (Kuster, 1851), *Thanatophilus rugosus* (Linnaeus, 1758)*, Thanatophilus sinuatus* (Fabricius, 1775); Staphylinidae: *Anotylus inustus* (Gravenhorst, 1806), *Atheta*
*(Atheta) castanoptera* (Mannerheim, 1830), *Creophilus maxillosus*** Linnaeus, 1758, *Omalium asturicum*** Fauvel, 1900, *Ontholestes murinus* Linnaeus, 1758, *Philontus* (*Philontus*) *varians* (Paykull, 1789); and unidentified species of families Fanniidae, Phoridae, Scathophagidae, Sciaridae, and Sphaeroceridae.

Mites (1533 specimens) were detached directly from the bodies of their host insects. These last mites belonged to 9 families, 16 genera, and 24 species: Ascidae: *Arctoseius semiscissus* (Berlese, 1892) **;* Digamasellidae: *Dendrolaelaps* sp.; Eviphididae: *Alliphis necrophilus* Christie, 1983, *Alloseius pratensis** (Karg, 1965), *Crassicheles holsaticus* Willmann, 1937, *Scarabaspis inexpectatus* (Oudemans, 1903)*;* Halolaelapidae: *Halolaelaps aeronauta*** (Vitzthum, 1918), *Halolaelaps octoclavatus* (Vitzthum, 1920), a Halolaelapidae sp.; Laelapidae: *Cosmolaelaps vacua* (Michael, 1891)*;* Macrochelidae: *Macrocheles glaber* (Müller, 1860), *Macrocheles merdarius* (Berlese, 1889), *Macrocheles muscaedomesticae* (Scopoli, 1772), *Macrocheles punctoscutatus*^*^ Evans & Browning, 1956; Nenteriidae: *Nenteria breviunguiculata** (Willmann, 1949), *Nenteria ritzemai** (Oudemans, 1903)*;* Parasitidae: *Cornigamasus lunaris* (Berlese, 1882), *Gamasodes spiniger* (Trägårdh, 1910), *Parasitus coleoptratorum* (Linneaus, 1758), *Parasitus fimetorum* (Berlese, 1903), *Poecilochirus austroasiaticus* Vitzthum, 1930, *Poecilochirus carabi* G. & R. Canestrini, 1882, *Poecilochirus subterraneus* (Müller, 1859)*;* Uropodidae: *Uropoda orbicularis*** (Müller, 1776). In addition, the presence of astigmatids was recorded on 52 occasions, but they were not quantified or identified.

Of the 48 species reported, 5 of them were recorded for the first time in the Iberian Peninsula (*) and 10 for the first time in the Foral Community of Navarra (**).

### Interaction Network

Each association established between a mite and an insect was quantified, regardless of whether the phoretic mite shared the host with others. Of the 1585 interactions established, 93.8% took place with beetles and 6.2% with flies (Figure 1 and Figure 2). Each mite species established an average of 1.3 links. The mean number of mites present on a host was 5.1; all of them could or could not be of the same species, and in the negative cases, they had different degrees of specificity with their hosts.

The interaction network of phoretic mites–host insects presents the following characteristics: It is composed of 1 shared compartment and 3 individual compartments. Each compartment is a subset formed by species of mites and insects that interact with each other, sharing species reciprocally, and that do not occur with the rest of the subsets. The first compartment is composed of 21 species of Mesostigmata, plus the order Astigmata, and 27 species of insects hosting more than one species. The second, third, and fourth compartments are individual compartments formed by a single mite species and a single host insect: Halolaelapidae sp. with *Geotrupes (G.) mutator*, *Macrocheles punctoscutatus* with Phoridae, and *A. semiscissus* with family Sciaridae (Figure 1).The asymmetry of the interaction network is 0.09, rescaled to [–1,1] according to Blüthgen *et al.* (2007) [61]. The asymmetry value is positive, close to 0, which means that there is a balance between the number of mite species (25) and the number of host species (30), although there is a very slight trend in favor of the hosts. This result indicates that in this network each species of mite corresponds to a host species, although in a few cases there is more than one.The interaction uniformity value (H_2_’) (which measures the specialization of the network) is 0.86. This value indicates a general specialization in the associations established between certain species of mites and their host insects (network values close to one mean a specialized network).

In total, 1585 interactions were established, although not all species contributed equally: *P. subterraneus* contributed 39.2%, *H. octoclavatus* 30.9%, *C. holsaticus* 7.3%, *P. austroasiaticus* 4.6%, *H. aeronauta* 4%, Astigmata 3.3%, *M. merdarius* 2.3%, Halolaelapidae sp. 1.9%, *N. breviunguiculata* 1.8%, *M. muscaedomesticae* 1.5%, *P. carabi* 0.8%, *M. glaber* 0.6%, *P. coleoptratorum* 0.4%, *S. inexpectatus* 0.4%, *U. orbicularis* 0.4%, *Dendrolaelaps* sp. 0.2%, and the species *A. necrophilus*, *A. pratensis*, *A. semiscissus*, *C. lunaris*, *C. vacua*, *G. spiniger*, *M. punctoscutatus*, *N. ritzemai*, and *P. fimetorum*, 0.1% each one.

Data about those interactions are detailed in Figure 1 and Figure 2, and Table 1.

*Poecilochirus subterraneus* appeared in 20.9% of the host insects collected. Each insect carried on average 9.9 ± 1.2 (range 1–126) mites located on the inside of the epipleuras and under the elytra of Silphidae and Scarabaeidae. Six hundred and twenty-one adult females were associated with individuals belonging to four species: *N. interruptus* (85.8%), *N. vespilloides* (13.2%), *O. (P.) coenobita* (0.6%), and *Th. ruficornis* (0.3%) (Figure 1 and Figure 2). Its host diversity was low, perhaps because it shows high specificity with *N. interruptus* (Table 1). 

*Halolaelaps octoclavatus* was removed from 8.3% of insects. Each one carried on average 18.8 ± 1.4 (range 1–63) mites, which travelled on the inside of the epipleuras and under the elytra of Histeridae and Silphidae. Four hundred eighty-nine deutonymphs established associations with beetles of four species: *H. unicolor* (5.3%), *M. brunneus* (94.3%), *N. interruptus* (0.2%), and *Th. rugosus* (0.2%) (Figure 1 and Figure 2). This species showed very low diversity (Table 1), perhaps due to its tendency to be associated *M. brunneus*. The dominance of relationships with this beetle determined a high specificity value. It is one of the Mesostigmata species with more than one host species that presented the highest specificity value (0.86). Its interactions with Silphidae remained anecdotal. 

*Crassicheles holsaticus* was the most frequent species, found on 30.1% of insects. Insects carried on average 1.2 ± 0.1 (range 1–4) mites, using their chelicerae to attach to the ventral abdomen surfaces of Scathophagidae and Staphylinidae. One-hundred and fifteen deutonymphs interacted with individuals of five species: *A. inustus* (1.7%), *A. (A.) castanoptera* (94.8%), *C. maxillosus* (0.9%), *O. asturicum* (1.7%), and flies of the family Scathophagidae (0.9%) (Figure 1 and Figure 2). Its biodiversity was low because of an almost total dominance of relationships with the species *A. (A.) castanoptera*, and it presented the highest specificity value (Table 1). 

*Poecilochirus austroasiaticus* was removed from 9.9% of insects. Each one carried on average 2.4 ± 0.2 (range 1–16) mites located inside the epipleuras and under the elytra of Silphidae. Seventy-three deutonymphs were associated with three species with *Th. ruficornis* (8.2%), *Th. rugosus* (8.2%), and *Th. sinuatus* (83.6%) (Figure 1 and Figure 2). It had a low diversity of host insects but high specificity with *Th. sinuatus* (Table 1).

*Halolaelaps aeronauta* was on a small number of insects (1.6%). Each insect carried on average 12.6 ± 0.8 (range 5–24) mites. Sixty-three deutonymphs were attached with their chelicerae to the ventral abdomens of Fanniidae (61.9%), Muscidae (7.9%), and Scathophagidae (30.2%) (Figure 1 and Figure 2). This species had medium diversity of host species (Table 1), showing preference for the family Fanniidae. 

*Macrocheles merdarius* was attached to 6.1% of host insects. Each insect carried 1.9 ± 0.2 (range 1–14) mites, which were attached by their chelicerae under the elytra on Histeridae and Silphidae, and between the head–thorax and thorax–abdomen junctions on Scarabaeidae. Thirty-six adult females were associated with five species of beetles: *M. brunneus* (5.6%), *N. interruptus* (5.6%), *O. (P.) coenobita* (11.1%), *Th. rugosus* (13.8%), and *Th. sinuatus* (63.9%) (Figure 1 and Figure 2). Its biodiversity and specificity (Table 1) mean a medium diversity of hosts and relatively specific associations with them. The number of interactions with *Th. sinuatus* was slightly higher than average.

Halolaelapidae sp. Thirty deutonymphs were attached among the pilosity of the ventral part of one specimen of *G. (G.) mutator*. This mite was the only one using this beetle for its dispersal. This species could be considered specific to *G. (G.) mutator*.

*Nenteria breviunguiculata* was carried by 4.5% of the host insects. Each carried 2.1 ± 0.2 (range 1–7) mites attached by an anal pedicel to the legs and abdomen of Histeridae, and to the abdomen of Staphylinidae. Twenty-nine deutonymphs interacted with *C. maxillosus* (13.8%), *M. brunneus* (3.4%), *O. murinus* (3.4%), and *P. (P.) varians* (79.3%) (Figure 1 and Figure 2). Its low biodiversity and high specificity values (Table 1) were due to the dominance of relationships established with *P. (P.) varians*.

*Macrocheles muscaedomesticae* was associated with 5.8% of the host insects. Each carried 1.3 ± 0.1 (range 1–3) mites. They were attached to the chelicera to the ventral part of the abdomen on Calliphoridae, Fanniidae, and Muscidae; and between the coxae of the legs of Scarabaeidae, Histeridae, and Silphidae. Twenty-three adult females interacted with *C. vomitoria* (4.3%), *Ch. albiceps* (4.3%), *Ch. megacephala* (4.3%), *Hydrotaea* sp. (4.3%), *M. brunneus* (4.3%), *Neomyia* sp. (8.7%), *O. (P.) coenobita* (8.7%), *Th. sinuatus* (21.7%), Fanniidae (4.3%), and Muscidae (34.8%) (Figure 1 and Figure 2). This mite had the highest diversity of host species and the lowest degree of specificity (Table 1).

*Poecilochirus carabi* was found on 1.6% of the host insects collected. Each host insect carried 2.6 ± 0.2 (range 1–9) mites, attached to the legs, the upper part of the thorax, and under the elytra of silphid beetles. Thirteen deutonymphs interacted 92.3% with *N. interruptus* and 7.7% with *Th. sinuatus* (Figure 1 and Figure 2). The value of specificity was high, and its host diversity was low (Table 1) because interactions with *N. interruptus* predominated. Its association with *Th. sinuatus* could be considered anecdotal or accidental. 

*Macrocheles glaber* was found on 2.6% of the host insects. Every one of them carried 1.1 ± 0.1 (range 1–2) mites. They attached with their chelicerae to the ventral part of the abdomen of Calliphoridae flies, and between the metathorax and the abdomen on Staphylinidae beetles. Nine adult females were found associated with *L. silvarum* (11.1%), *O. (P.) coenobita* (77.8%), and *Th. rugosus* (11.1%) (Figure 1 and Figure 2). Its specificity value was 0.79 and had medium diversity (0.42). Its interactions showed slight dominance of *O. (P.) coenobita* over the others, which shows high specificity.

*Parasitus coleoptratorum* was phoretic on 1.9% of the host insects. Each host insect carried 1.2 ± 0.1 (range 1–2) mites, which were attached under the elytra for beetles and between the metathorax and the abdomen for flies. Seven deutonymphs established associations with *O. (P.) coenobita* and individuals of Muscidae (Figure 2): 85.7% of them were on beetles and the 14.3% on Muscidae flies (Figure 1). This species had low host diversity (Table 1); however, it associated with them with high specificity, especially *with O. (P.) coenobita*. 

*Scarabaspis inexpectatus* was found on 1.3% of the host insects. They carried 1.5 ± 0.1 (range 1–2) mites per insect, which were attached to Scarabaeidae around the perimeter of the head–prothorax and prothorax–mesothorax junction on its ventral part. Two adult females, three adult males, and a deutonymph interacted only with individuals of the species *Onthophagus (P.) coenobita*, giving it the highest specificity value (1).

Next, 1.6% of the host insects carried *Uropoda orbicularis*. Each host carried 1.2 ± 0.1 (range 1–2) mites. This species travelled on the elytra of Histeridae and Scarabaeidae using its anal pedicels. Six deutonymphs interacted with *H. unicolor*, *M. brunneus*, and *O. (P.) coenobita*. Based on its values of biodiversity and specificity (Table 1), no predominant relationships were detected with anyone host.

*Dendrolaelaps* sp. was found in 0.64% of the host insects collected. Each one of them carried on average of 1.5 ± 0.1 (range 1–2) mites, under the elytra. Histeridae were the carriers. Three deutonymphs interacted with *M. brunneus* (33.3%) and *S. (S.) subnitescens* (66.7%) (Figure 1 and Figure 2). It associated with both insect species equally and with high specificity (Table 1).

The mesostigmatid species *A. necrophilus*, *A. pratensis*, *A. semiscissus*, *C. lunaris*, *C. vacua*, *G. spiniger*, *M. punctoscutatus*, *N. ritzemai*, and *P. fimetorum* interacted with 0.32% of the hosts insects collected. The host species and their locations were as follows:

One adult female of *Alliphis necrophilus* under the elytra of *N. interruptus*; one adult female of *Alloseius pratensis* inside the epipleura of *N. interruptus*; one adult female of *Arctoseius semiscissus* in the ventral part of the abdomen of Sciaridae (in the interaction matrix, this species formed an independent compartment because it was the only species found that used these flies for its dispersal, and at the same time, these flies are its unique hosts); one deutonymph of *Cornigamasus lunaris* in the ventral part of the abdomen of Sphaeroceridae; one adult female of *Cosmolaelaps vacua* under the elytra of *S. (S.) subnitescens*; one deutonymph of *Gamasodes spiniger* in the ventral part of the abdomen of Sphaeroceridae; one adult female of *Macrocheles punctoscutatus* in the ventral part of the abdomen of Phoridae (this species was another of those that formed an independent compartment in the interaction matrix because it was the only species found that used these flies for its dispersal, and at the same time, these flies are its unique hosts); one deutonymph of *Nenteria ritzemai* over the elytra of *M. brunneus*; one deutonymph of *Parasitus fimetorum* under the elytra of *O. (P.) coenobita*.

In these species, values of diversity and specificity were 0 (one mite captured) and 1 (on one insect), which shows that in this study their phoretic associations were unique and completely specific.

The Astigmata hypopus stage was detected in 16.7% of the host insects. As only their presence was taken into consideration, it is not possible to give an estimate of mean numbers or the range of astigmatids carried by the hosts. Hypopus appeared on 52 occasions (each presence is equivalent to an interaction) in number of different host insects: *D. frischii* (3.8%), *D. undulatus* (1.9%), *Hydrotaea* sp. (1.9%), *N. interruptus* (23.1%), *N. vespilloides* (11.5%), *O. (P.) coenobita* (1.9%), *P. (P.) varians* (2.8%), *S. (S.) subnitescens* (1.9%), *Th. rugosus* (7.7%), *Th. sinuatus* (17.3%), and families Muscidae (23.1%) and Sphaeroceridae (1.9%) (Figure 1 and 2). In general, they were located on the abdomens of flies and Staphylinidae, and under the elytra of the rest of the beetles. This order showed high diversity of host insects and a low degree of specificity (Table 1). If these mites had been identified at a lower taxonomic level, differences in diversity and specificity values would have been detected.

From the results of this work, and after a thorough review, the following new associations between insects and phoretic mites emerged:Astigmata mites interact with myriapods, crustaceans, and insects; we newly report the presence of hypopal stages on Dermestidae (*D. frischii* and *D. undulatus*), Geotrupidae (*G. (G.) mutator*), Histeridae (*H. unicolor*, *M. brunneus* and *S. (S.) subnitescens*), Scarabaeidae (*O. (P.) coenobita*), Silphidae (*N. interruptus*, *N. vespilloides*, *Th. ruficornis*, *Th. rugosus* and *Th. sinuatus*), Calliphoridae (*C. vomitoria*, *Ch. albiceps*, *Ch. megacephala* and *L. silvarum*), Fanniidae, Muscidae, and Scathophagidae.*M. muscaedomesticae* interacts with the largest number of insect species. Its newly reported host species are *Ch. albiceps*, *M. brunneus*, *Neomyia* sp., *O. (P.) coenobita*, and *Th. sinuatus*.*C. holsaticus* was detected for the first time in association with Scathophagidae.*M. merdarius* has *M. brunneus*, *N. interruptus*, *O. (P.) coenobita*, *Th. rugosus*, and *Th. sinuatus* as newly identified hosts.The host insects of *N. breviunguiculata* was hitherto unknown. This is the first record of this Uropodina being associated with *C. maxillosus*, *P. (P.) varians*, *M. brunneus*, and *O. murinus*.Deutonymphs of *P. subterraneus* now have *O. (P.) coenobita* and *Th. ruficornis* among their list of hosts.*P. austroasiaticus* has *Th. sinuatus* as a new host with high specificity.*M. glaber* has *L. silvarum* and *Th. rugosus* as new hosts.*U. orbicularis*—previously cited as a generalist phoretic mite on coprophilic and saprophylic insects—is for the first time cited as phoretic on *M. brunneus*.The genus *Dendrolaelaps* joins *M. brunneus* and *S. (S.) subnitescens* (Histeridae) among the previously reported hosts.*P. coleoptratorum* shows high specificity in this region and uses Muscidae flies and *O. (P.) coenobita* as hosts.*A. necrophilus* interacted uniquely and specifically with *N. interruptus*.*A. pratensis* was for the first time found on *N. interruptus* (Silphidae); it was also found (in the same trap) on its usual hosts (*O. (P.) coenobita* and *Hister* spp.), which could indicate that this species switches hosts in the trap to survive.The new record of *A. semiscissus* females on Sciaridae supports the specificity of the relationship.There are few records of *C. vacua* as a phoretic mite; it is newly reported to live on *S. (S.) subnitescens*.The interactions established by *M. punctoscutatus* with Phoridae flies, *N. ritzemai* with *M. brunneus*, and Halolaelapidae sp. with *G. (G.) mutator*, are new phoretic sites for these species.Deutonymphs of *P. fimetorum* are specialists for *O. (P.) coenobita*.*S. inexpectatus* had *O. (P.) coenobita* as a unique host.

## 4. Discussion

During the sampling period, mites were mainly associated with beetles, perhaps because they tend to inhabit the soil, like mites and unlike flies, so the probability of interacting with them is higher [30,62,63]. A group of mites (*A. semiscissus*, *C. lunaris*, *G. spiniger*, and *M. punctoscutatus*) was only associated with flies; Astigmatids, *C. holsaticus*, *H. aeronauta*, *M. glaber*, *M. muscaedomesticae*, and *P. coleoptratorum* used both flies and beetles; and the remining 15 species were exclusively related to beetles for their transport.

Interactions between these phoretic mites and host insects exhibited an interaction network classified as specialist [57], although polyxenic species are more numerous than monoxenic species. Polyxenic species commonly live in ephemeral or temporary habitats. As such, in terms of the survival of the species, it is more important for them to find new habitats to colonize than to find insect species for their transport [64]. The polyxenic mite in this study established predominated relationships with certain host species (except those of *M. muscaedomesticae* and astigmatids). These dominances, together with the high number of species (11 species, 73%) of absolute exclusivity, were the causes of this specialized network.

Species such as *S. inexpectatus*, *C. holsaticus*, and *H. octoclavatus* were the most specialized species, and *M. muscaedomesticae* was the most generalist species. These same species showed lower diversity values for specialists and higher for generalists. Biodiversity and specificity indexes are inversely proportional. As such, a given mite presents less specificity for its host by using a greater number of host species. In general, species of this study had lower values of diversity and higher specificity than expected.

***Remarks****:* The numbers of mites that remained attached to the bodies of their hosts may be influenced by their attachment mechanisms, the mobility of mites during travel, and their location in the hosts’ bodies. *P. carabi* involved a small number of attached mites in their phoretic dispersal. It is a highly active mite [65] and extremely mobile; it can leave its carriers temporarily or permanently [66,67] and attach to the host using its pretarsal leg claws. This mechanism is apparently less secure than using chelicerae or anal pedicels. In addition, this species travels on exposed areas of its hosts, such as legs, head, and thorax [68,69]. Other species of the same family, such as *P. coleoptratorum*, *P. fimetorum*, *P. austroasiaticus*, and *P. subterraneus* use their claws, but these species have higher numbers of attached individuals. However, the four forementioned species travel on more protected areas, for example, under the elytra [70], which reduces accidental detachment during flight or when they enter the galleries on the ground [69].

Host specificity and its attachment location on the host’s body are unique characteristics of Mesostigmata mites [12,69,71]. The fact that each species or stage of the same species selects a specific location on the host insect’s body, sharing them or not with other mite species, could serve to reduce intra or interspecific spatial competition [25]. The individuals belonging to the same species or stage are in the same places and in aggregates [72].

Host body size is considered the most influential factor in selection by mites [73]. A larger body tends to encourage selection. This is certainly an important factor for mite dispersion, but it is not the only one. The number of mites in the host should be balanced [74], maximizing the use of the host surface but allowing its locomotion, especially flight [24]. Many mites could hinder movement [49], slowing or even preventing relocation to a new resource [75]. At the same time, the number of mites travelling on the same host will be determined by the size of both, host and mites. Larger species, such as *P. carabi*, had low numbers of dispersive individuals per host (2.6), whereas smaller species, such as *P. subterraneus* or *H. octoclavatus*, had higher numbers of dispersive individuals (9.9 and 18.8, respectively).

We want to highlight the novelty of this study. It presents the first interaction network between phoretic mites and their host insects in a Mediterranean region and the first time that an interaction network applies to decomposing animal samples of forensic interest. We can conclude that the degree of interaction specificity between phoretic mites and a host insect is a biological and ecological characteristic of the mites at the time of association. We reveal the importance of various factors on the effectiveness of mite dispersal, such as the body size of the host and mite, the numbers of mites and insects in terms of availability, location on the host’s body, mobility during the transport, and the attachment mechanism of phoretic mites.

## Figures and Tables

**Figure 1 insects-13-00610-f001:**
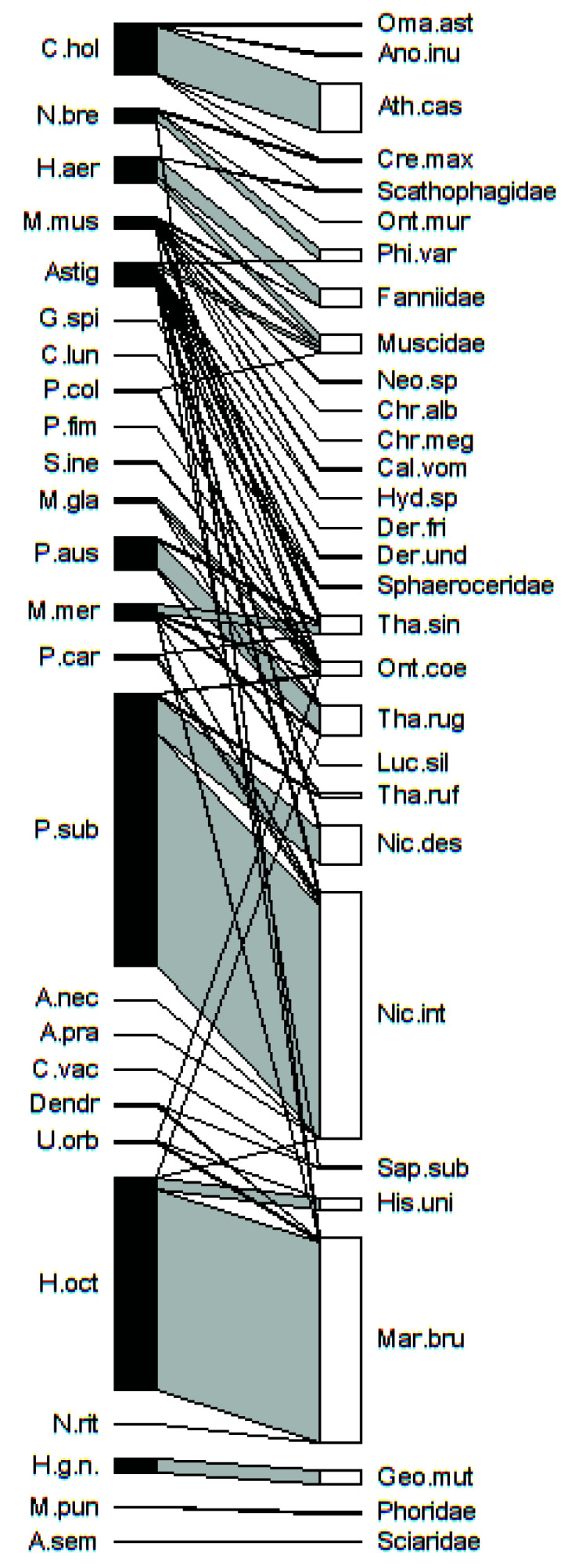
Bipartite network of interaction between phoretic mites’ species (left/black) and families/species of host insects (right/white). The thickness of the bond (grey) indicates the number of interactions observed. The abbreviations used are the following: *A. necrophilus* (A.nec), *A. pratensis* (A.pra), *A. semiscissus* (A.sem), *A. inustus* (Ano.inu), *Astigmata* (Astig), *A*. (*A*.) *castanoptera* (Ath.cas), *C. holsaticus* (C.hol), *C. lunaris* (C.lun), *C. vacua* (C.vac), *C. vomitoria* (Cal.vom), *Ch. albiceps* (Chr.alb), *Ch. megacephala* (Chr.meg), *C. maxillosus* (Cre.max), *Dendrolaelaps* sp. (Dendr), *D. frischii* (Der.fri), *D. undulatus* (Der.und), *G. spiniger* (G.spi), *G*. (*Geotrupes*) *mutator* (Geo.mut), *Halolaelapidae* sp. (H.g.n.), *H. aeronauta* (H.aer), *H. octoclavatus* (H.oct), *H. unicolor* (His.uni), *Hydrotaea* sp. (Hyd.sp), *L. silvarum* (Luc.sil), *M. glaber* (M.gla), *M. merdarius* (M.mer), *M. muscaedomesticae* (M.mus), *M. puntoscuatus* (M.pun), *M. brunneus* (Mar.bru), *N. breviunguiculata* (N.bre), *N. ritzemai* (N.rit), *Neomyia* sp. (Neo.sp), *N. vespilloides* (Nic.des), *N. interruptus* (Nic.int), *O. asturicum* (Oma.ast), *O*. (*P.*) *coenobita* (Ont.coe), *O. murinus* (Ont.mur), *P. austroasiaticus* (P.aus), P. *carabi* (P.car), *P. coleoptratorum* (P.col), *P. fimetorum* (P.fim), *P. subterraneus* (P.sub), *P*. (*P.*) *varians* (Phi.var), *S. inexpectatus* (S.ine), *S*. (*S.*) *subnitescens* (Sap.sub), *Th. ruficornis* (Tha.ruf), *Th. rugosus* (Tha.rug), *Th. sinuatus* (Tha.sin), and *U. orbicularis* (U.orb).

**Figure 2 insects-13-00610-f002:**
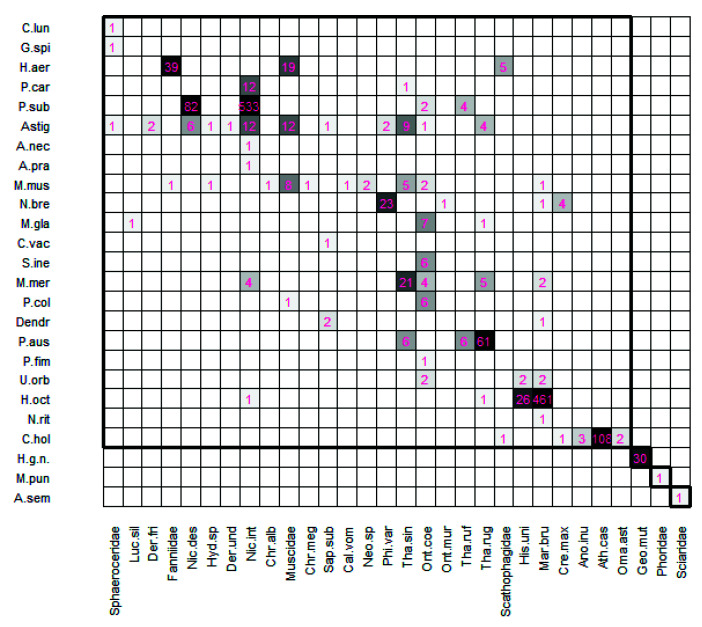
Interaction matrix between mite species (rows) and host insect families/species (columns). The intensity of the color of the grid is directly proportional to the number of interactions (purple number inside the square). The abbreviations used are the same as in Figure 1.

**Table 1 insects-13-00610-t001:** Values of total number of host insects collected (N) for each species, number of host species for each species (S), result of the biodiversity index (BI) and result of the specificity index (SI). (*) Appearances, not number of mites. The abbreviations used are the same as in Figure 1.

Mite sp.	Astig	N.bre	N.rit	U.orb	C.lun	G.spi	P.col	P.fim	P.aus	P.car	P.sub	Dendr
N	52 *	29	1	6	1	1	7	1	73	13	621	3
S	12	4	1	3	1	1	2	1	3	2	4	2
BI	0.85	0.36	0	0.8	0	0	0.29	0	0.29	0.15	0.25	0.67
SI	0.36	0.8	1	0.56	1	1	0.86	1	0.83	0.92	0.86	0.74
**H.aer**	**H.oct**	**H.g.n.**	**A.nec**	**A.pra**	**C.hol**	**S.ine**	**M.gla**	**M.mer**	**M.mus**	**M.pun**	**A.sem**	**A.vac**
63	489	30	1	1	115	6	9	36	23	1	1	1
3	4	1	1	1	5	1	3	5	10	1	1	1
0.53	0.11	0	0	0	0.1	0	0.42	0.57	0.84	0	0	0
0.68	0.94	1	1	1	0.94	1	0.79	0.61	0.41	1	1	1

## Data Availability

Data is contained within the article.

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
