# Peer review of "First Interaction Network of Sarcosaprophagous Fauna (Acari and Insecta) Associated with Animal Remains in a Mediterranean Region (Northern Spain)"

_insects, 2022, doi:10.3390/insects13070610_

Round 1
Reviewer 1 Report
The main question addressed by the research is the first interaction network analysis between phoretic mites and their host insect applicate to animal remains of forensic relevance. This is an interesting forensic paper with new information of organisms that may be relevant for dating Post –Morten Intervals. The work is ok as it is in regard to methodology. The conclusions consistent with the evidence and arguments presented and they address the main question posed.
Some additional comments :
Line 54: Please indicate that PMI means Post-Morten Interval.
Lines 235 - 241: Names of species must be in italics.
Line 418: It seems that the word 'habitats' should be at the end of the sentence "Polyxenic species commonly live in ephemeral or temporary."
Author Response
Line 54: Please indicate that PMI means Post-Morten Interval. Done, Thanks.
Lines 235 - 241: Names of species must be in italics. Done, Thanks.
Line 418: It seems that the word 'habitats' should be at the end of the sentence "Polyxenic species commonly live in ephemeral or temporary." Yes. Done, Thanks.
Reviewer 2 Report
The paper outlines the relationship between carrion feeding insects and their mites. The observations and recording of new species in the Navarra area will be of interest to entomologists. The authors start with the premise of using these associations for forensic purposes and estimates of time since death or the post mortem relocation of a body, however the authors have not demonstrated the the data supports this application, the most useful insects will be those that arrive early but the mites appear to be mostly associated with beetles that typically arrive at the later stages of decomposition . The observations are of ecological interest and the introduction could be re-worked to take more of an ecological focus. Some additional background on the life cycle of these mites that would help put the observations of the stages detected in context. The development of the interaction network is interesting highlighting the polyxenic nature of the interactions. The authors highlight some instances that have been described as "unique and completely specific" however these are based on a single observation so care needs to be taken in making any inferences about specificity. The methods are generally clear but some additional information about the habitat would be useful, the authors mention the area is green all year but is this grassland/woodland? they give general climate information but some more detail could be included (average rainfall, summer/winter temperatures) as habitat preferences of the insects and their mites might be important. The methodology suggest data were collected on the number of mites their sex, stage, position on their host and attachment mechanism these results could be explored in more detail, some of these are mentioned in the discussion but data to support their discussion is not presented in the results.
Grammatical issues and typos:
line 29: applicated - applied
line 29: Also reveals.... - The data reveals
line 32: was collected - were collected
line 45: being insects and mites are the - with insects and mites being the
line 53: or removal - delete
line 83: there is a slight growing tendency - there is a move
line 105: mixt - mixture
line 112: Each infected insects were - Infected insects were
Author Response
Some additional background on the life cycle of these mites that would help put the observations of the stages detected in context. Done. We added some information. Thanks.
The authors highlight some instances that have been described as "unique and completely specific" however these are based on a single observation so care needs to be taken in making any inferences about specificity. We have cleared that this is in this study. Thanks.
The methods are generally clear but some additional information about the habitat would be useful, the authors mention the area is green all year but is this grassland/woodland? they give general climate information but some more detail could be included (average rainfall, summer/winter temperatures) as habitat preferences of the insects and their mites might be important. These informations are in reference 34, published by us in this journal. It is indicated in the text. Thanks.
The methodology suggest data were collected on the number of mites their sex, stage, position on their host and attachment mechanism these results could be explored in more detail, some of these are mentioned in the discussion but data to support their discussion is not presented in the results. The information was presented in the results although we clarified the information for a better understanding. Following I give you an example. Thanks.
Crassicheles holsaticus was the most frequent species, found on 30.1% of insects. Insects carried on average 1.2 ± 0.1 (range 1-4) mites, using their chelicerae to attach to the ventral abdomen surface of Scatophagidae and Staphylinidae. Hundred fifteen deutonymphs (previously as number of interations) interacted with individuals of 5 species:....
Grammatical issues and typos:
line 29: applicated - applied Done. Thanks.
line 29: Also reveals.... - The data reveals Done. Thanks.
line 32: was collected - were collected Not done. Fauna, although the word is often treated as plural, it’s traditionally a singular noun encompassing a collection of things (https://grammarist.com/usage/flora-fauna/) . Thanks.
line 45: being insects and mites are the - with insects and mites being the Done. Thanks.
line 53: or removal - delete Done. Thanks.
line 83: there is a slight growing tendency - there is a move Done. Thanks.
line 105: mixt - mixture Done. Thanks.
line 112: Each infected insects were - Infected insects were Done. Thanks.
Reviewer 3 Report
Reviewer comments
Manuscript title: First Interaction Network of Sarcosaprophagous Fauna (Acari and Insecta) Associated with Animal Remains in a Mediterranean region (Northern Spain).
The manuscript by Pérez-Martínez and Moraza documents the relationship between the phoretic mites and their host sarcosaprophagous insects in the Mediterranean region. The life history of phoretic mites is unique, and their species composition, sex ratio and the occurrence of hypopus have the potential value in the investigation of criminal cases. So this study has its merits to be published. However, some revisions are still needed.
Firstly, I don’t think the title is appropriate. Before I read the text, what I get from the title is that this study used animal carcasses, but the fact is the author only used pork bait. Secondly, this study is only a species survey and did not involve the information regarding the arrival and departure of insects and mites, so I wondered why the author emphasized the importance of faunal succession in the abstract and introduction. Thirdly, introduction used forensic entomology as the background, implying that this study could be used in the detection of criminal homicide cases. However, the value of the results to the death case was not reflected in the results or discussion.
Other specific comments and recommendations are:
Line 103-104: Three traps were used in this study to sample insects and mites. But the readers were not familiar with them. After I searched the relevant references, I found an article wrote by the authors of this study - Gamasina Mites (Acari: Mesostigmata) Associatedwith Animal Remains in the Mediterranean Region of Navarra (Northern Spain). I suggest the authors cite the reference here or give a simply introduce about the traps.
Line 106: EAFE recommended use pig carcasses as the substitute of human corpses in body decomposition and insect succession study. The pig baits were not recommended by the EAFE. So this sentence is not necessary.
Lin 107-110: I guess the propylene glycol was used as preservative in the traps. Please specific.
Line 118: Please give the references to the taxonomic keys of mites.
Line 157-175: Authority name of each species should be added.
Line 188: Change “individual” to “individuals”.
Line 191: Change “mesostigmatids” to “mesostigmata”
Line 235-241: The species name should be in italics.
Author Response
Line 103-104: Three traps were used in this study..... I suggest the authors cite the reference here or give a simply introduce about the traps. The reference it is cited but we have clarified the sentence to a better understanding. Thanks.
Line 106: EAFE recommended use pig carcasses as the substitute of human corpses in body decomposition and insect succession study. The pig baits were not recommended by the EAFE. So this sentence is not necessary. Deleted. Thanks.
Lin 107-110: I guess the propylene glycol was used as preservative in the traps. Please specific. It is specified in lines 110-111. Thanks.
Line 118: Please give the references to the taxonomic keys of mites. Done. Thanks.
Line 157-175: Authority name of each species should be added. Done. Thanks.
Line 188: Change “individual” to “individuals”. The phrase was changed to better understanding. Thanks.
Line 191: Change “mesostigmatids” to “mesostigmata” Done. Thanks.
Line 235-241: The species name should be in italics. Done. Thanks.